# The enduring pursuit of public science at U.S. land-grant universities

**Bradford Barham[1]☯, Jeremy Foltz[iD][1]☯*, Ana Paula Melo[2]☯**

**1** Department of Agricultural and Applied Economics, University of Wisconsin-Madison, Madison, Wisconsin, United States of America, **2** Department of Economics, Howard University, Washington, DC, United States of America

☯ These authors contributed equally to this work.
* jdfoltz@wisc.edu

**Data Availability Statement:** All relevant data are within the manuscript and its Supporting information files.

**Funding:** JF and BB: AFRI [grant no. 2016-AFRI-005843-A1641] from the USDA National Institute of Food and Agriculture, URL: NIFA.USDA.GOV/

## Abstract

Since the 1990s, universities have faced a push toward output commercialization that has been seen as a potential threat to the public science model. Much less attention has been given to the enduring nature of internal organizational features in academia and how they shape the pursuit of traditional scholarly activities. This article exploits four waves of representative, random-sample survey evidence from agricultural and life science faculty at the 52 major U.S. land-grant universities, spanning 1989-2015, to examine faculty attitudes/preferences, tenure and promotion criteria, output, and funding sources. Our findings demonstrate that faculty attitudes toward scientific research have remained remarkably stable over twenty-five years in strongly favoring intrinsic and public science goals over commercial or extrinsic goals. We also demonstrate the faculty's positive attitudes toward science, an increased pressure to publish in top journals and secure increasingly competitive grants, as well as declining time for science. These trends suggest a reconsideration of university commercialization strategies and a recommitment of universities and their state and federal funders toward fostering public agricultural and life science research.

## Introduction

Much ado has been made about potential changes in faculty scholarly activities at public universities in the United States [1–3] and globally [4–7]. Some authors raise concerns about external incentives that might pull faculty away from public science toward commercialization and intellectual property (IP) right creation and protection [8–11]. Others warn of similar effects from changing internal incentives such as erosion of tenure and promotion of performance-based budget models at universities [12]. A third common line of inquiry questions whether the pursuit of IP and refereed articles are substitutes or complements in the academy, with solid evidence to suggest that for basic science research they are largely complements [13–16]

This article examines how a broader set of scholarly activities have evolved over time at all 52 original 1862 U.S. Land-grant universities (LGUs) using representative random-sample survey data from agricultural and life science faculty, spanning 26 years from 1989 to 2015. It

The funders had no role in study design, data collection and analysis, decision to publish, or preparation of the manuscript. JF: USDA-Hatch (WIS01853) grant through the University of Wisconsin-Madison. URL nifa.usda.gov. The funders had no role in study design, data collection and analysis, decision to publish, or preparation of the manuscript.

**Competing interests:** The authors have declared that no competing interests exist.

situates this analysis by exploring first how much internal and external incentives really changed at U.S. public universities over the last quarter century. Specifically, it documents the evolution of two key internal factors (faculty research preferences, and tenure and promotion incentives) and a largely external one (funding sources for faculty research). In terms of scholarly activity, the analysis incorporates both traditional scholarship efforts, such as peer-reviewed articles, training undergraduates, graduate students and 'post-docs' in scientific methods, and extension/outreach activities aimed at public stakeholders, as well as measures of commercialization activity (including patents, licensing revenues, and start-ups). Attention to this broader activity mix extends previous work that mostly focused on research outputs and changes in the direction of research [3, 15, 17].

Major institutional pull factors [7] are often linked to potential changes in incentives and faculty activities. Key pull factors include the 1981 U.S. Bayh-Dole Act and similar legislation elsewhere, rapid growth and expansion of university technology-transfer offices, legal rulings in support of patenting life sciences [16] and other types of information-science based innovations. Push factors [18] emanate from fiscal constraints and corresponding incentive changes, especially at U.S. publicly funded universities. Specifically, cuts in federal and state spending on higher education, higher operating costs of research, and tuition freezes drove universities toward the pursuit of efforts to expand innovation, participate in start-ups, and/or pursue alternative funding streams or performance-based budgeting schemes [19]. These shifts in opportunities and constraints have led to lots of policy discussions about the potential implications for faculty scholarly activities.

For three decades, agricultural and life science faculty at U.S. LGUs have been at ground zero of these changes [20, 21]. In the 1990s, a major wave of biotechnology research efforts associated with genetic and genomic oriented science dramatically altered the potential for innovation and commercialization opportunities. Longstanding engagement with industry, or agribusiness and biotech firms, via extension systems and sponsored research positioned LGUs well for pivoting toward a more commercial approach to doing science [22, 23]. Major investments were made in technology transfer offices and academic patenting efforts. And, budget cuts came from many angles, including declines in state general-purpose funds, tuition caps, declining USDA research funds, the end of 'pork barrel' programs, and cuts in federal and county-level support for Cooperative Extension activities [19, 24].

Nonetheless, exclusive focus on these external 'forces of change' overlooks the internal organizational logic of public universities that have long shaped faculty activity choices. Especially but by no means solely, U.S. LGU faculty have enjoyed and defended formal and informal mechanisms of ensuring academic freedom, shared governance, and peer-based tenure and promotion (T&P) reviews. Also potentially overlooked is the critical role of incentives for faculty from peer-reviewed extramural grant competitions (mostly from public funds) that demand a track record of success publishing articles, training students, and often generating broader social impacts as well. Accounting for the stability of these mechanisms and incentives, and the premium they place on traditional scholarly activities, helps to explain the deep resilience/stability of scholarly attitudes and activities we find over the last quarter century.

We feature in Fig 1 a simple diagram of these internal and external factors that likely shape scholarly activity. Individual choice models in social sciences assume that agents make choices based on preferences, endowments (or income), and constraints. In an academic setting, faculty preferences rest on the foundation of academic freedom—the choice of what to research and teach is a deeply cherished and long championed principle of work at public universities. Academic freedom is also part of what attracts high quality scientists to academia as opposed to other more lucrative career options. Individual choice models are also shaped by 'incentives' or tradeoffs; prices in consumer choice models, or more generally opportunity costs that

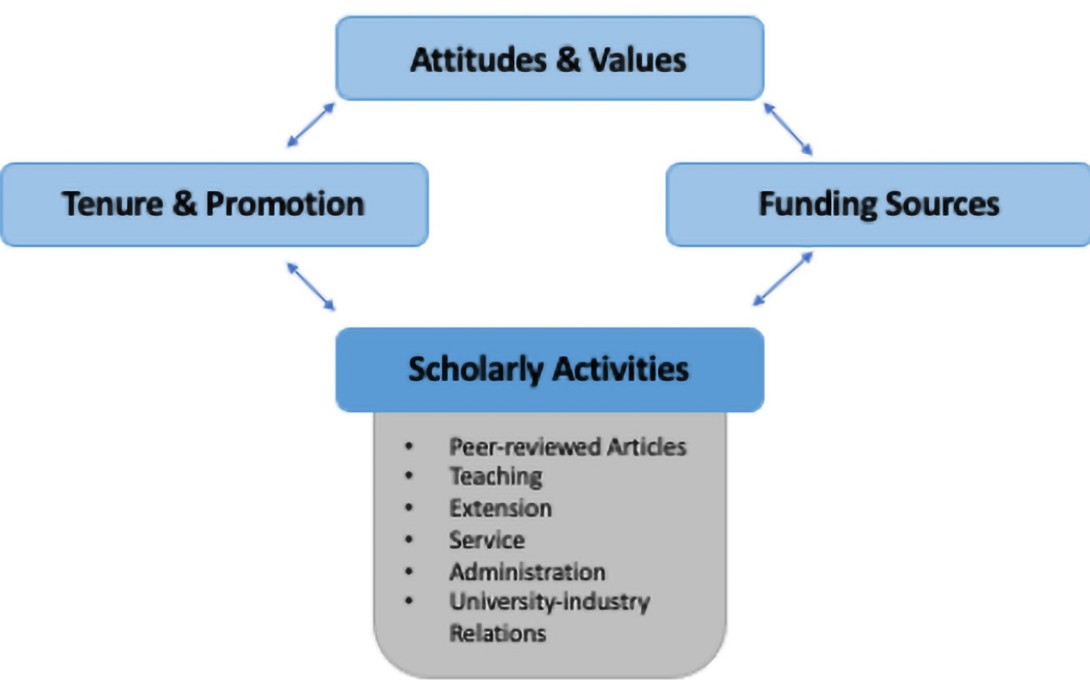

**Fig 1. Internal factors shaping scholarly activity.** Diagram of internal and external factors shaping scholarly activity: attitudes and values, tenure and promotion, an funding sources.

agents face when comparing alternatives, whether they be purchases or monetary or time allocation choices. Correspondingly, scholarly activities are driven by interlinked incentives from 'tenure and promotion', faculty's own 'attitudes and values', and 'funding sources' which serve to finance the research activities and summer salaries of faculty.

Attitudes and values sit in the center of Fig 1 with tenure and promotion incentives on one side and grant funding sources on the other. Accordingly, tenure and promotion (T&P) criteria represent the primary internal incentives or tradeoffs faculty face when making choices over what activities to pursue and how to use their work time. In this individual choice model of faculty activity, access to research funding is also highly valued for both purposes. Accordingly, the sources and incentives associated with grantsmanship provide incentives that shape faculty activity choices.

These three factors—attitudes and values, tenure and promotion, and funding sources—sit above the scholarly activity box as the main internal and external drivers explored here, with causal mechanisms potentially flowing in multiple directions. For example, success in published articles could help to generate more success in external grant competitions, which could, in turn, lead faculty to value publishing more articles. These circular feedback loops both make for a resilient system and a challenging undertaking to pursue statistical identification of the inter-relationships portrayed here. We do not attempt to 'identify' the causes. We do, however, show the strong association or correspondence between these factors and scholarly activities over time at 'ground-zero' of where the purported push of traditional scholarly activity toward more commercialization could have occurred.

In our empirical analysis, we test for changes over time and across faculty in attitudes and values, the incentive effects of tenure and promotion, and funding sources. We relate these changes to the evolution of scholarly activities as shown in Fig 1. The surveys provide

consistent questions across a 26 year period for these factors and activities that we use to examine patterns of change.

The evidence displays remarkably stable and resilient faculty attitudes and support for incentive systems consistent with the open, public science model of traditional scholarly activities (specifically research focused on peer-reviewed articles and books, instruction and training, and extension). Intrinsic attitudes (e.g., joy of science) appear to play a major role in driving research choices and support for open scientific inquiry over commercialization, with some evidence of heterogeneity in attitudes across basic versus applied research orientations.

Historical comparisons of faculty perceptions of actual and preferred T&P criteria reinforce the stability of science-driven attitudes. It illustrates the enduring strength of incentives that reward peer-reviewed publications and instruction evaluations over all other T&P criteria. Faculty research funding continues to rest primarily on public funds and traditional sponsored research sources (commodity organizations and industry collaborations) rather than on opportunities for 'commercialization'. In turn, public research funding and industry sponsorship may have become even more contingent on success in producing traditional scholarly outputs, especially peer-reviewed articles.

Not surprisingly, continuity in the mix of scholarly activities and self-reported time use reflect the stability seen in the three factors just discussed. In fact, in the two more recent surveys, we find an increase in the average number of refereed published articles, a steady level of training graduate students, and an increase in the training of post-docs. We also find similar time spent on undergraduate instruction, more time spent on the pursuit of extramural competitive grants and administrative duties, along with relatively low and recently declining levels of recent engagement with commercialization activities. The closing arguments in the Discussion section reflect on the implications of our findings for how scientists, university leaders, and policymakers might approach the continuing pursuit of public science in research-oriented universities.

## Attitudes & values

In order to test changes in attitudes and values, we report survey evidence on faculty responses to factors they perceive as shaping research problem choice. The questions use a Likert Scale of 1–5, with 5 highest and 1 lowest. Factor analysis shows that faculty answers to the survey questions divide into two sets of factors, which we label as 'intrinsic' and 'extrinsic' motivations (see Methods and materials section for details).

Over the past quarter century, agricultural and life sciences faculty responses strongly and consistently favored intrinsic over extrinsic factors, as is found elsewhere in the literature [25]. For example, 'enjoy doing this kind of research' or 'scientific curiosity' average 4.46 and 4.14 on the Likert scale across the years of data. These two intrinsic scores (Fig 2) are almost two points higher than the averages of 2.33 and 2.82 for the extrinsic factors marketability of final product and request by clientele. A minor level of heterogeneity across faculty is revealed when we divide the sample into two groups of basic and applied research orientation. The basic group scores somewhat higher on intrinsic attitudes and considerably lower on extrinsic attitudes than applied scientists (S1 Fig). Nonetheless, the applied scientists average Likert scores still strongly favor intrinsic attitudes over external ones as critical to shaping research problem choice.

Although not shown in Fig 2, faculty are also strongly motivated by the 'importance to society' of their work, giving it an average of 4.05, comparable to but slightly lower than the intrinsic items, and significantly above the extrinsic ones. In the factor analysis, importance to society is similarly correlated with both the intrinsic or extrinsic factors, leaving it arguably as

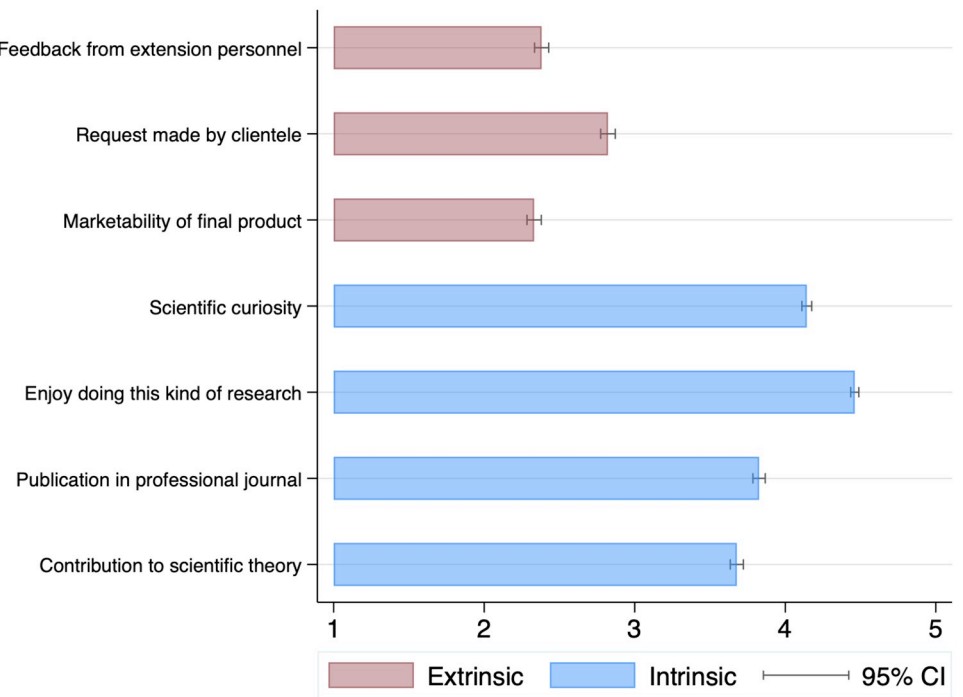

**Fig 2. Average per motivation item, from 1989 to 2015.** Figure displays cross-year average per item, with 95 percent confidence intervals. N = 3,047.

its own independent motivation (see Table 4). This response closely aligns with a public science model, where faculty preference for creating knowledge transfers for innovation that broadly serves society cut across intrinsic and extrinsic motives.

Fig 3 shows that while faculty preferences across intrinsic and extrinsic factors have varied a bit across the past quarter century, overall the patterns remain remarkably stable. Intrinsic motivations dominate with most faculty having values at the top of the index. Meanwhile, there is a more even distribution of extrinsic values with most faculty consistently below 3 on a Likert scale, with only a few showing high extrinsic motivation. It is worth underscoring the remarkable stability of faculty responses that identify the joy of science and scientific curiosity as the lead factors shaping research choice, rather than extrinsic motivations. This emphasis is robust to all of the observed heterogeneity we tested (including tenure status, gender, field or discipline, and as mentioned above, basic versus applied research orientation).

## Tenure & promotion

The survey data provide two types of T&P criteria questions: 1) faculty perceptions of the factors influencing T&P decisions at their universities; and 2) faculty preferences on what type of T&P criteria they would privilege if given the choice. We believe that perceptions of faculty on the rules for tenure and promotion represent a better measure of the incentive effects of the rules than a close reading of the rules themselves, since many campuses have unwritten parts of the rules that are well understood incentives for faculty. For the four time periods of measures, we report results for the ranking of perceived T&P criteria and a score for how the survey respondents—if they could—would adjust up or down the 'rank' they gave to the current perceived T&P criteria.

Fig 4 shows the yearly ranking. For each tenure and promotion criteria, respondents chose a score reflecting their perceived degree of importance for each item. The size of the circles

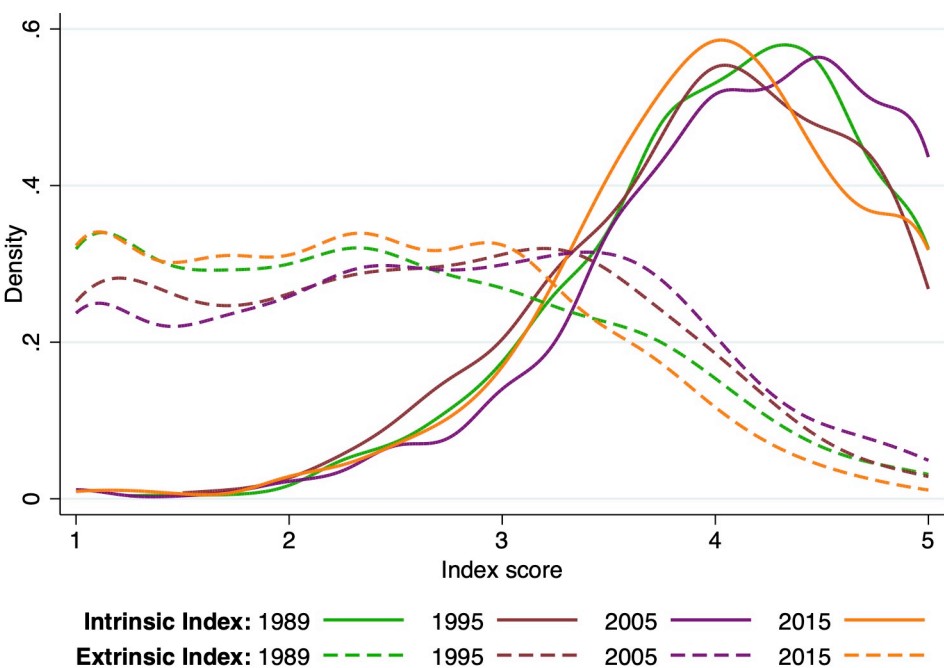

**Fig 3. Distribution of attitudes and values indexes, by year.** This figure displays the distribution of the two indexes by year. The solid lines correspond to the intrinsic index and the dashed line corresponds to the extrinsic index. To construct the indexes, we averaged the items scores within individual and within each grouping resulting from the factor analysis to create an index. More details can be found in the Materials and methods section. N = 3,047.

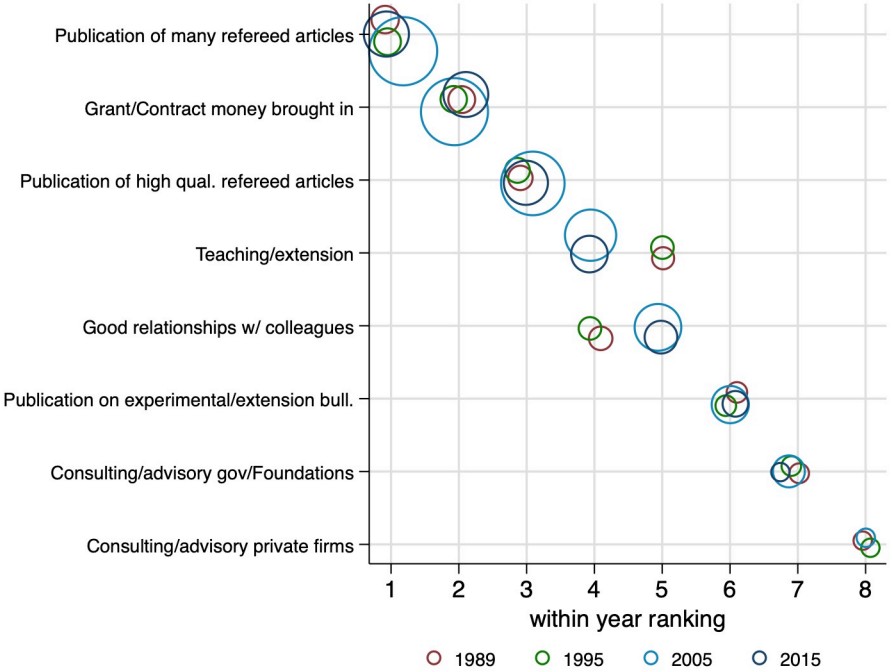

**Fig 4. Faculty perceptions of T&P criteria.** This figure displays the criteria ranking by year. The circles are weighted by the within-year average per item, with larger circles reflecting larger within-year averages. N = 3,045.

reflects the within-year average of each criterion—bigger circles mean the average score for that specific criteria is higher relative to a smaller circle in each year. Similarly to preferences regarding research choice, faculty perceptions of incentives from T&P criteria have been remarkably consistent over the past quarter-century with no material changes or variations in these rankings (only teaching and 'good relations' flip places). The within-year average scores—reflected in the graph by the size of the circles—reveal an increase in the perceived importance of the higher-ranked criteria, such as publications of many and high-quality papers, and grant and contract money, in 2005 and 2015.

The top four T&P criteria in faculty perceptions of their university's rules are all traditional scholarly activities: quantity of research articles, quality of research articles, grant and contract money brought in, and teaching or extension evaluations. In contrast, the bottom two T&P criteria as perceived by faculty are commercially oriented activities: consulting or advisory work, with private firms, foundations, or government firms. Good relations with colleagues falls in the middle of the ranking.

These faculty perceptions of T&P criteria highlight that faculty perceive major incentives to produce traditional scholarly outputs. They perceive much lower incentive to engage in commercial or clientele oriented activities, although they may at times be complements rather than substitutes to scientific output [15, 20].

Faculty preferences for how they would change T&P criteria (Fig 5) suggest that they would prefer to place an even higher relative value on public science activities but not on commercially oriented ones. For example, the two top T&P criteria they would increase in importance are publication of high quality research and teaching or extension evaluations. The other T&P criteria they would substantively move up in the rankings are contributions to department and

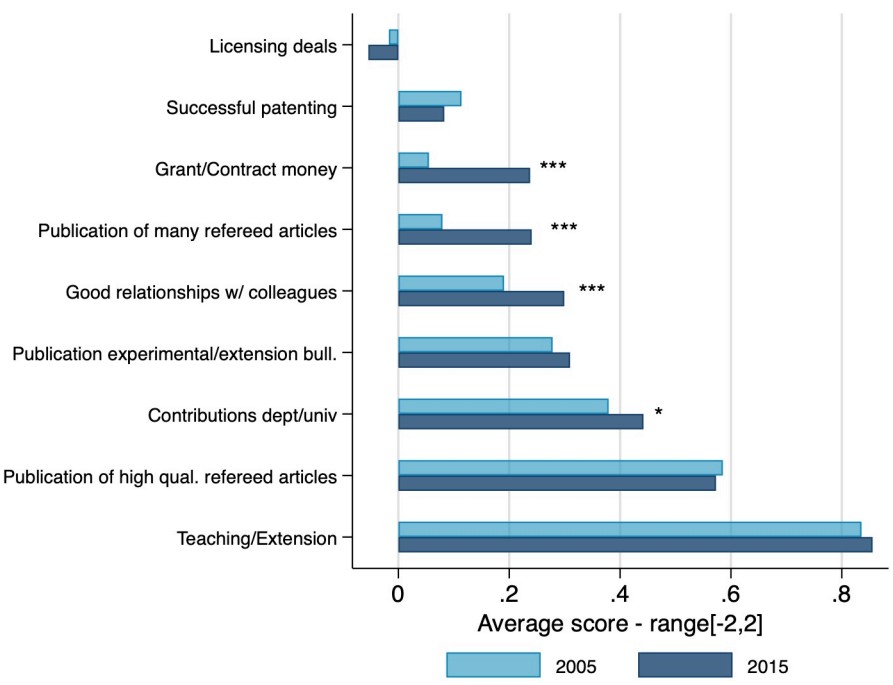

**Fig 5. Faculty preferences with respect to T&P criteria.** Figure displays average scores per preference criteria. Responses range from -2 (Much Less Weight) to +2 (Much More Weight). Statistical t-tests between 2005 and 2015 data within a category are shown in the graph. *: 10%, **: 5%, ***:1%. N = 1,772.

university and publication of extension bulletins, both of which are also consistent with support for rewarding service and public science activities.

Both perceived and preferred criteria for T&P strongly favor the pursuit of excellence in peer-reviewed research, teaching, and service—the longstanding basis for evaluation at LGUs—over externally oriented commercial activities. This is also consistent with the attitudes and values expressed by faculty that privilege scientific values over commercial ones.

## Funding sources

Grant and contract funding is an essential input to the pursuit of science and graduate training among professors at LGUs and research universities. In the tenure and promotion analysis, funding is also perceived by LGU faculty to be a critical output with respect to T&P criteria. Here to explain the funding context for faculty, we report on the main sources of research funding over time and demonstrate the continued predominance of public money and sponsored research contracts over commercial activities.

As shown in Fig 6, in 2005 and 2015, public funding sources provide the majority of funds reported by LGU faculty. In both years, federal and state funding accounted for almost 60 percent of faculty research funding, with university funding adding another 11–12 percent to public sources of research funds. Combined, sponsored research from private industry and commodity organizations accounted for about 16–17 percent, while funding from patent royalties and licenses accounted for less than 0.3 percent. Most of these percentages did not change significantly between 2005 and 2015, with the exception of state grants (- 2 p.p), foundations (+ 1.5 p.p), and others (- 0.8 p.p). This temporal stability in faculty reported research

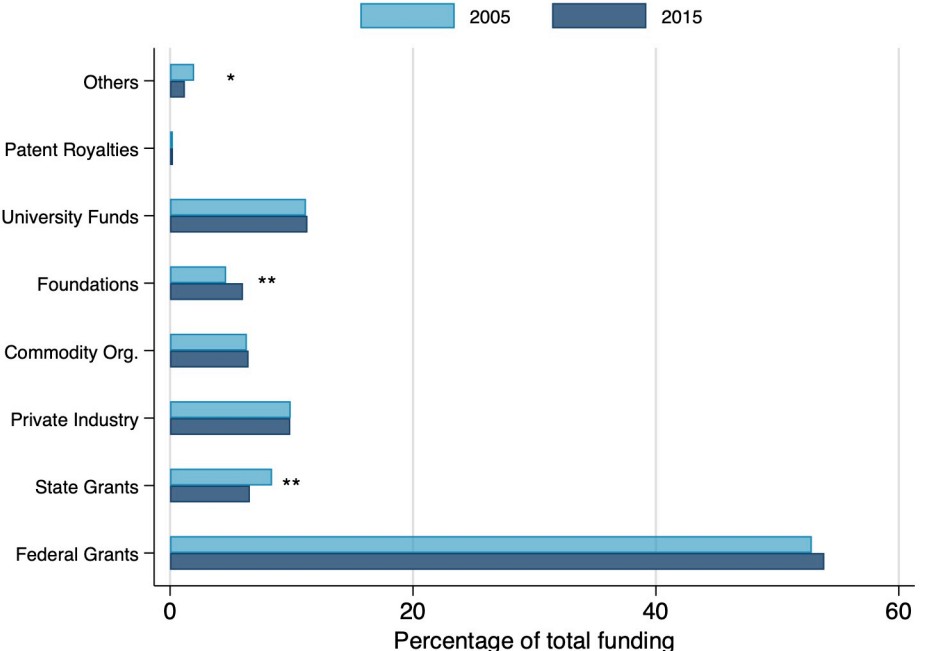

**Fig 6. Distribution of total funding by source, in 2005 and 2015.** Figure displays average responses for percent of funding received by each source. Average total funding—in 2005: $162,897, in 2015: $301,759. Federal grants include answers for the percentage of total funding received from: Experiment Station funds (Hatch and McIntire-Stennis), USDA competitive grants, USDA cooperative agreements, National Science Foundation (NSF), National Institutes of Health (NIH), Department of Energy (DOE), and Other federal government agencies. Statistical t-tests testing difference between years are shown in the graph. *: 10%, **: 5%, ***:1%. N = 1,787.

grant funding is consistent with NSF data on total U.S. research and development funding for life and agricultural sciences which shows a steady role of federal funding since the mid-1980s to the present (Sources: Survey of Research and Development Expenditures at Universities and Colleges 1972–2009; Higher Education Research and Development (HERD), 2010-current).

Other analyses of these data show a notable synergy [26], or positive correlation, between respondents success in securing federal grants and sponsored research funds (akin to the 'star scientists' discussed in [27]. Among highly funded LGU scientists, they are almost invariably attracting both higher levels of public agency grant funding, along with sponsored research funds. Far fewer of those star scientists are generating much for their labs from commercialization.

Finally, it is worth pointing out that the competitive nature of federal grant seeking has increased in recent years for LGU scientists [23, 24]. As shown in Fig 7, there has been a significant shift toward more competitive peer-reviewed grants (USDA, NSF, NIH) and away from Experiment Station Funds and USDA Cooperative Agreements. In 2015, competitive grants accounted for 51 percent of the federal total, as compared to 42 percent in 2005. Coincident with this change has been a dramatic increase between 2005 and 2015 in the average faculty receipt of grant funding among the top 20 LGUs (148 percent increase), according to the US News and World Report ranking, relative to faculty in non-top 20 LGUs (33 percent increase). While our work shows no significant change in the distribution of funding sources, top 20 LGU faculty moved from parity in average funding levels with non-top 20 faculty to having 70 percent more research funding on average. This suggests a growing inequality in conditions between faculty at the top LGUs and their peers at lower ranked LGUs.

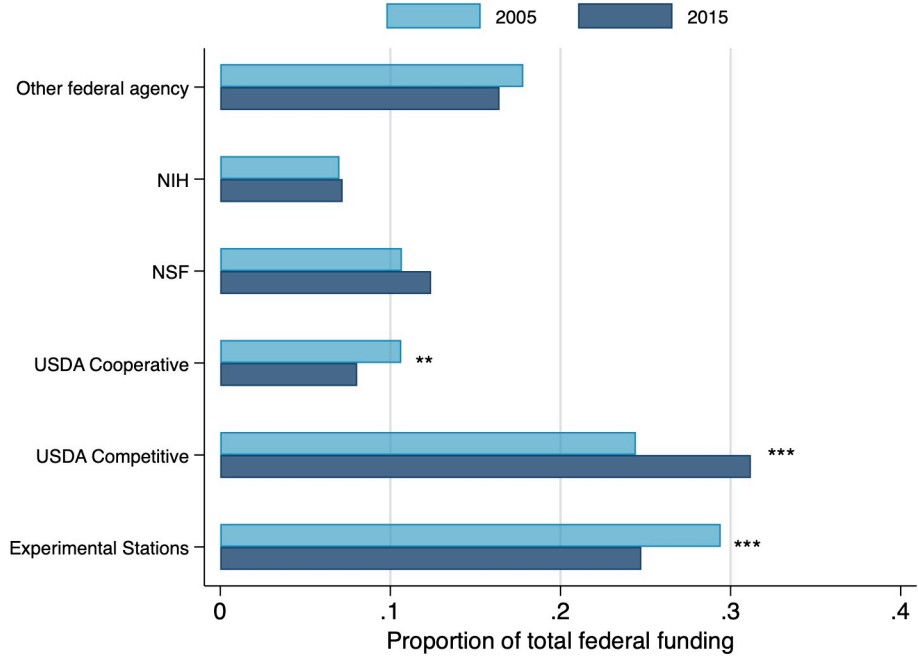

**Fig 7. Distribution of federal funding, by type of agency, in 2005 and 2015.** Figure displays federal funding by type. Percents are calculated over total federal funding. Federal funds correspond, on average, to 50 percent of total funding. Average total funding—in 2005: $162,896, in 2015: $301,759. Statistical t-tests are showed in the graph. *: 10%, **: 5%, ***:1%. N = 1,787.

This shift toward competitive grants also privileges peer-reviewed publications as a driver for grant competition success relative to experiment station and cooperative agreement funding. The latter tend to be based more on administrative criteria or 'pork-barrel' arrangements. A growing role of competitive grants, and the continued reliance on public sources, is likely to fuel faculty pursuit of traditional scholarly outputs over commercialization ones. Competitive peer-reviewed grants more generally reward "playing safe" [28], unless large monetary incentives compensate for the switching costs [29]. Success in competitive peer-reviewed public grants depends fundamentally on the design of the proposed research and the reputation of the faculty members pursuing the grant, which in turn will be heavily determined by their previous and current production of scholarly articles [30] and to a lesser extent by training of students and post-docs.

## Scholarly activities

We close the empirical presentation by reviewing the scholarly activities of faculty to explore how they align with the evidence on attitudes and values, tenure and promotion criteria, and funding incentives. We focus on peer-reviewed journal articles, graduate student, and post-doctoral scholars as outputs and on the time devoted to research, teaching, extension, and administration as evidence of other less measurable activities and as a reflection of the varied demands faculty face.

Faculty research productivity, measured by the average number of refereed journal articles over the previous five years, is reported in Fig 8. The first three observations, 1989, 1995, and 2005 are very similar (12.2–13.3 range), but are about 25 percent lower than the 2015 average of 15.7. There are some degrees of heterogeneity in comparisons across the basic versus applied research orientation groups, but this gap closes over time (See S1 Table). The stability and more recent growth of faculty research article productivity (discussed again below) is

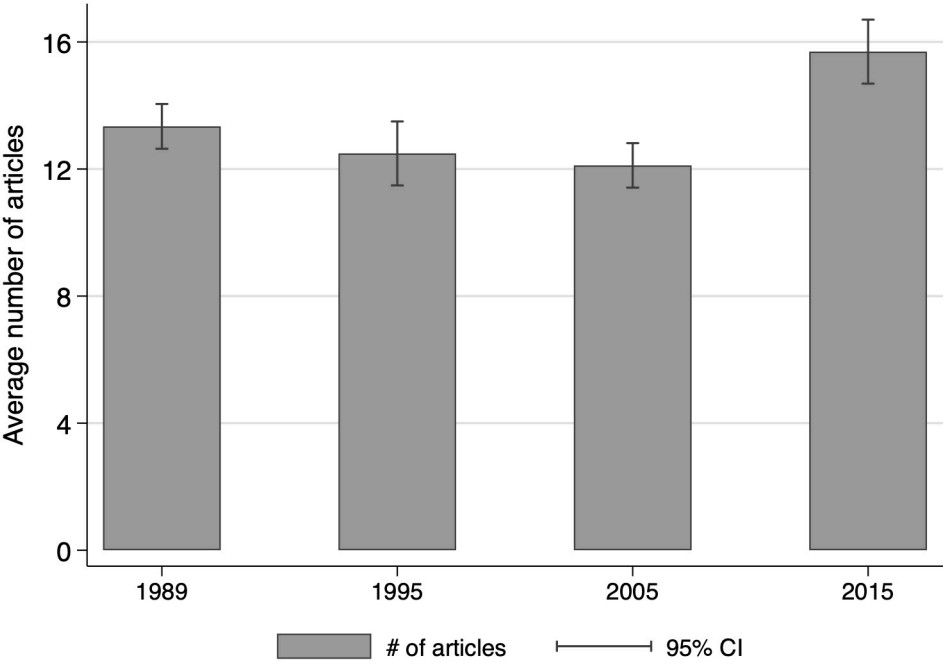

**Fig 8. Average number of articles published, by year.** This figure displays the number of total articles published in the 5 years before the survey. N = 3,014.

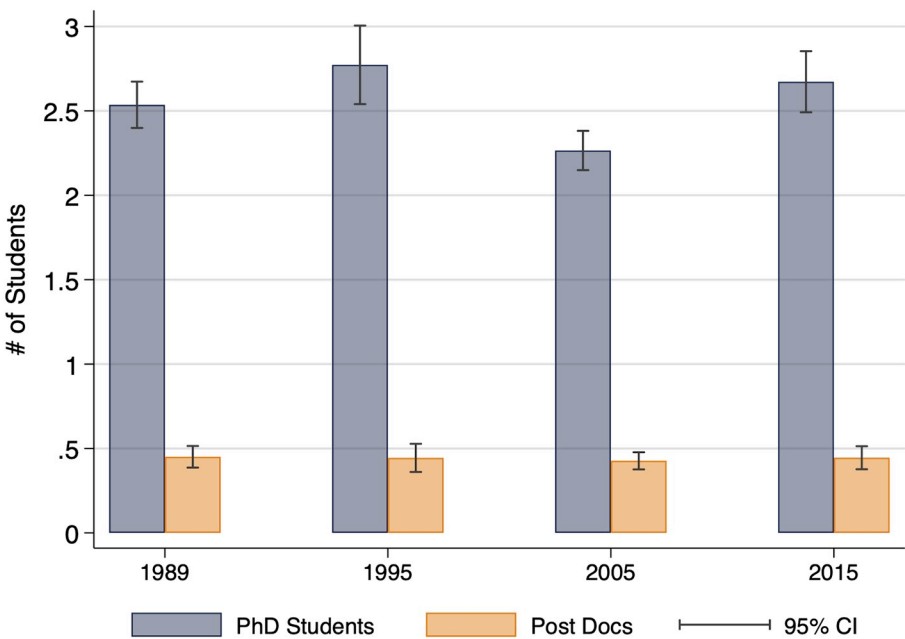

**Fig 9. Average number of PhD and post-doctoral students under supervision, by year.** This figure displays the average number of students currently under supervision. N = 2,991.

consistent with the enduring importance and stability of faculty preferences, T&P incentives, and grant sources.

Graduate instructional activity is shown in Fig 9, specifically the number of PhD students and post-docs for which faculty served as the primary advisor over the previous five years. Despite a statistically significant drop in PhD students in 2005, which rebounded in 2015, these figures are remarkably similar across the quarter century. Consistent with research article production, and the secondary, but continued importance, of instruction in T&P, there is no sign of decline in faculty emphasis on graduate and post-doctoral training.

Time use trends of faculty are reported in Fig 10, as a proportion of total work hours, which for 2005 and 2015 was a median of 53 and 52 hours per week, respectively. The time allocation data show a steady decline in the proportion of time allotted to research from about 58 percent in 1989 to a low of 43 percent in 2015. S2 Fig reveals that in the last ten years, research time proportion held steady around 45 percent in the top 20 LGUs, while it declined from 48 percent to 40 percent in the non-top 20 LGUs. Combined with the growing gap in funding outcomes mentioned above, these results suggest a potential divergence in trends across different sub-samples of LGUs in terms of work conditions. In particular, it suggests conditions to do research at top LGUs are being maintained or even improving, while they are stagnating or deteriorating at lower ranked LGUs. Such a "rich get richer" type of outcome is worthy of further research.

Teaching time, across the sample, held steady at around 28 percent, supporting the perception of faculty that teaching is a significant basis for evaluating T&P. By contrast, the proportion of time spent on administration/other activities grew from 9 to 16 percent, putting time for science among LGU faculty under increasing pressure [3, 31]. This increase was especially strong in the 2005–15 period for the non-top 20 LGUs. Given the singular importance of research article production in T&P evaluations and successful research grant acquisition, these

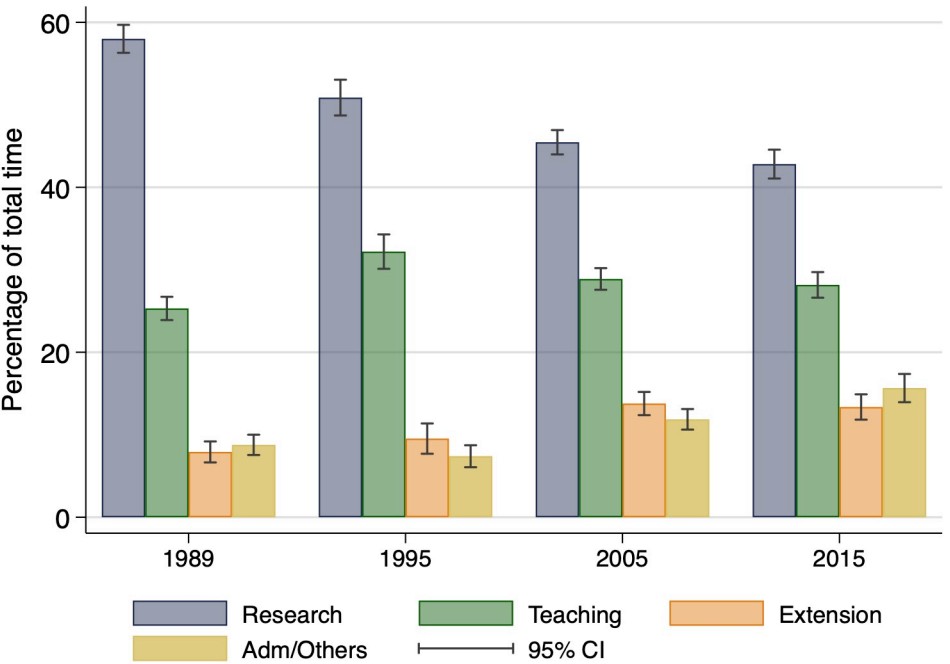

**Fig 10. Allocation of faculty research time by type of activity.** This figure displays the percent of time *actually* allocated to each group of activities. This might differ from percentages defined in their formal appointments. N = 2,846.

declining time allocation trends for research are likely putting considerable pressure on faculty and graduate students and research staff as well to produce more with less time available [32, 33].

Moreover, a closer look at the division of time spent on research activities alone (Fig 11) shows in 2015 that administration of current grants and preparation of grants, accounted for about 40 percent of the time LGU faculty devoted to research, leaving only about 60 percent of research time for actual research. When combined with the estimated 43 percent of total weekly work time spent on research, we find that in 2015 LGU faculty had only about 26 percent of total work time for doing actual research.

This finding regarding limited time for science has two additional implications. One is that faculty research productivity appears to have risen in recent decades [34]—note the increase in faculty research output combined with less time for research. Some share of that, however, may be explained by an increase in collaborative research and or a proliferation of journals, rather than actual productivity increases [35, 36]. The other implication is that with increasing pressure on faculty to find time for research, which is essential to both T&P and competitive research funding opportunities, faculty may be reducing effort for less rewarded activities such as public engagement or commercialization. While research and commercialization can be synergistic activities, in that quality research can lead to commercialization, that extra effort for commercialization requires time and effort that is lacking at many LGUs.

These same LGU survey data allow us to track the recent evolution of faculty commercialization activities, reported for 2005 and 2015. Table 1 reveals low and flat or declining levels of commercial activities [26]. Only about one in eight LGU faculty made any disclosures or patents in both five-year time periods. Additionally, the proportion of LGU faculty with a patent decreased from 14 percent in 2005 to 11 percent in 2015. For those who did participate in

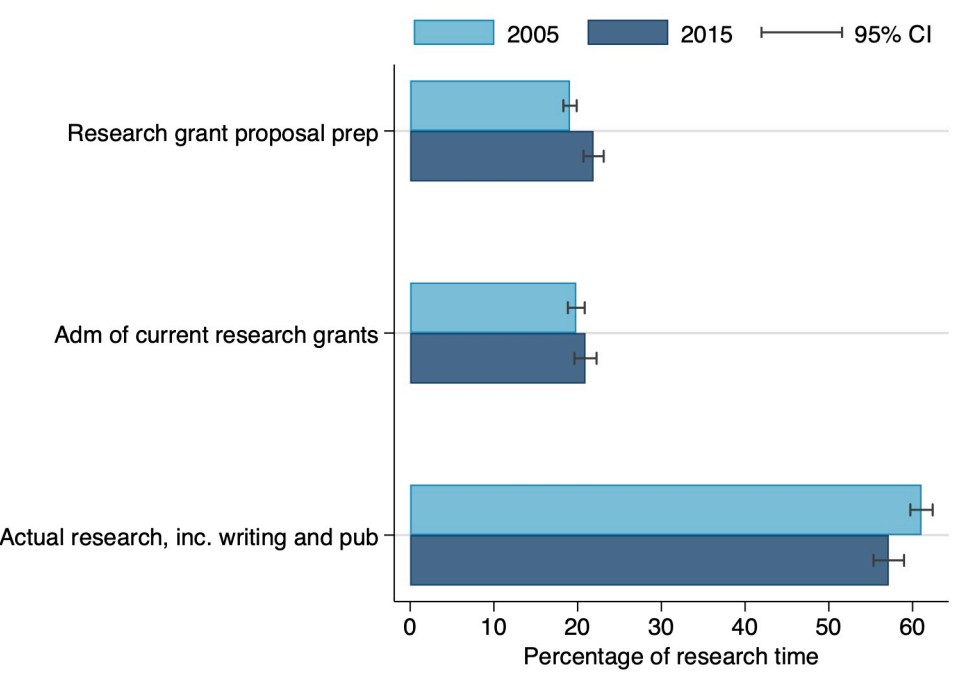

**Fig 11. Research time allocation across sub-activities, in 2005 and 2015.** This figure reports results restricted to 2005 and 2015 due to data availability. N = 1,741.

these activities, the number of invention disclosures and patents averaged around 2.4 over five years, with no significant differences across the two time periods. In terms of licensing revenues, only 4–5 percent of faculty reported any patent royalties in their labs. The average amounts for those receiving any patent royalties were $24,816 in 2015 and $16,470 in 2005, and those levels accounted on average for about 5 percent of total research funds for those receiving any at all. Participation in start-ups is also very uncommon.

These low commercialization participation outcomes follow directly from the evidence presented on attitudes, T&P criteria, major funding sources, and faculty time constraints. They also suggest that concerns with the impact of efforts to increase commercialization on

**Table 1. Evolution of faculty commercialization activities.**

|  | Invention Disclosure | | Patent Applications | |
|---|---|---|---|---|
|  | % faculty with any | # of inventions | % faculty with any | # of patents |
| 2005 | 13.51 | 2.59 | 13.93 | 2.40 |
| 2015 | 12.09 | 2.82 | 10.68 | 2.46 |
| p-value [2015- 2005] | 0.374 | 0.528 | **0.040** | 0.910 |
|  | Start-up companies founded | | Royalties income to lab | |
|  | % faculty with any | # of start-up | % faculty with any | % funding from royalties |
| 2005 | 3.71 | 1.68 | 4.44 | 5.18 |
| 2015 | 2.53 | 1.22 | 5.03 | 4.70 |
| p-value [2015- 2005] | 0.160 | 0.390 | 0.561 | 0.958 |

This table displays the faculty's commercialization activities by year. Number of inventions, patents, start-ups and percent of funding from royalties are calculated within individuals that indicated participation in any of these activities. P-values correspond to the t-test for mean difference between 2005 and 2015. N = 1,542 for Inventions Disclosure, Patent Apps, and Start-up companies. For Royalties, N = 1,801.

traditional scholarly activities may be overlooking the enduring nature of how faculty preferences, T&P criteria, and research funding opportunities shape actual faculty activities especially in public research university contexts, where time constraints and competitive grant pressures appear to be increasing.

## Discussion

Survey data spanning 25 years of faculty attitudes and activities at U.S. LGUs demonstrate an enduring pursuit of traditional academic scholarship that focuses on refereed journal article production, training of graduate students and post-doctoral scientists, and instructional and extension activities. Two factors internal to U.S. LGUs that appear key in shaping this outcome are (i) faculty preferences for research-problem choice based on intrinsic factors, and (ii) tenure and promotion incentives that highly reward articles and grants, along with instructional evaluations. The continued predominance of competitive, publicly funded research grants provides reinforcement for this emphasis on traditional scholarship activities. Placed against a context of considerable university promotion and external pressure to pursue new forms of commercial knowledge transfer and revenue generation, U.S. LGU agricultural and life science faculty appear to have, in fact, added a relatively low degree of academic commercialization activities to their overall activity set, one that previously included a high degree of academic engagement with industry and farm organizations [26]. These conclusions are backed up by supplementary information in S1 and S2 Figs, and S1 Table.

At a fine-grained level, the findings reveal that LGU faculty are under growing performance pressures that reinforce their current activity mix, consistent with experiences in other life science fields [37]. Securing public research funds has become increasingly competitive with declining award rates for NSF and NIH grants [38]. With lower hit rates on grant applications, faculty are likely to apply for more grants (assuming that they do not instead withdraw entirely from the race). Lower success rates also make the production of high quality and quantity of research articles more valuable to successful grantsmanship, which increases the pressure to get more research done. At the same time, rising administrative duties, and more time spent on compliance with research awards, collide with the time that faculty report having for doing actual science. Combined, these pressures almost certainly help to explain why faculty participation in commercialization activities and interest in doing so remain very low and even declined somewhat between 2005 and 2015.

We next discuss three potential limitations to these findings. First is whether these findings for agricultural and life science faculty at U.S. LGUs are representative of the attitudes, incentives, and activities of other faculty in major research universities. Meaningful differences could certainly emerge based on fields of knowledge [5] and on variations in the internal and external factors examined above. However, the 52 LGUs represented in this study include many top public research institutions (e.g., Cornell University, UC Berkeley, UC Davis, University of Illinois, University of Maryland, University of Minnesota, and University of Wisconsin), and the same T&P rules that privilege journal articles, grants, and teaching are very likely the same for many faculty in those universities, especially since most divisional committees (e.g., biological, physical, and social sciences) include faculty from colleges of agricultural and life sciences. Moreover, LGU agricultural and life science faculty also have a long history of active participation in university-industry relations that might make it a smoother transition for them to engage in more commercialization relative to traditional scholarship, which probably translates into more willingness to do so. And, the combination of applied sciences in agriculture and basic sciences centering on biological research may also mean that the LGU faculty surveyed were better situated in terms of the biotechnological opportunities relative to

other faculty at these universities. A reasonable working hypothesis for other U.S. public research university faculty is that, outside of medical schools, they would look similar to or even be less active in the non-traditional scholarly activities (such as commercialization) than the faculty surveyed for this study.

A second limitation is the potential for self-selection bias associated with participation in a voluntary survey. Notably, the response rates to the survey have declined somewhat over time (see the Materials and methods section), perhaps associated with the time pressures mentioned above, especially the squeeze on 'time for science'. However, it is hard to explain the remarkable stability of response scores to the questions asked and at the same time imagine the self-selection process that would generate that stability. Non-response bias tests administered on the 2015 data revealed no significant results for an array of standard possible sources (discipline, rank of faculty, gender).

A third limitation is that this study does not explore the causality of the mechanisms involved. It tests for changes over time and for significant differences within factors, but it does not attempt to explicitly test the direct links between preferences, tenure and promotion criteria, or funding with activity choices. To do so in a robust fashion would be a considerable challenge, one that would probably require either some type of natural experiment (in terms of exogenous policy shocks experienced in a subset of universities or disciplines) or a lab experiment exploring stated preferences. Most likely, the opportunity to study just one of the potential mechanisms, (e.g., tenure incentives on activity mix) would be the most one could hope for. In this instance, describing the comprehensive correspondence between a model of faculty decision-making that incorporates preferences, tenure and promotion incentives, and funding opportunities/constraints and then linking those to the choice of activities provides a more valuable contribution for a broad audience of scholars concerned about the ongoing pursuit of public science.

## Conclusion

This article uses survey data from 1989 to 2015 to reveal the remarkable stability and resilience of traditional academic scholarship among agricultural and life scientists at the 52 United States Land-Grant Universities, even in the face of considerable external pull and push forces to pursue commercialization. LGU faculty's enduring commitment to peer-reviewed research activities, especially articles and grants, as well as to graduate training and other instructional activities, appears to rest on a robust combination of attitudes (or preferences), tenure and promotion incentives, and increasingly competitive research grant awards, all of which privilege traditional academic scholarship, rather than commercialization. This finding stands in strong contrast to concerns regarding a major erosion in the public science model associated with academic commercialization or other budgetary pressures on public universities.

Perhaps more important is the value of the evidence spanning such a long time period for university faculty, staff, and administrators, legislatures at the state and federal level, and the citizens and other stakeholders asked to support public university activity. The main activities of the public science model remain central to motivating and guiding faculty activity, and public funding continues to be the main driver of faculty research and graduate training activities. Given these findings, the messaging for policy discourses could benefit from being clear that university faculty activities continue to strongly privilege science, training and instruction, over commercialization and intellectual property right pursuits. The viability and vitality of public universities are likely to be better served with that main message rather than one that urges and encourages faculty to pursue 'third pillar knowledge transfer activities' and promotes that heavily to other stakeholders. Those activities are better viewed as complements to

ongoing public science efforts, rather than substitutes. Shrinking public funding for U.S. land-grant university agricultural and life sciences will likely just mean less research and traditional scholarly activity and perversely perhaps even less commercialization efforts.

## Materials and methods

### Surveys of U.S. land-grant agricultural and life scientists

This work analyzes data from four waves of surveys conducted among tenure-track faculty in agricultural and life sciences departments across 52 Land Grant Universities in 1989, 1995, 2005 and 2015. Questions in all years were designed as a replication and extension of a 1979 survey created by [39]. This eliminates concerns with wording inconsistencies. All questions used in this study are identical to each other across years. For more details on the surveys, see [40–43]. The surveys were approved by the U. of Wisconsin-Madison Social and Behavioral Sciences Institutional Review Board SB-2015–0924. In line with the IRB approval, informed consent was obtained electronically from participants, but the signature requirement was waived.

The sample frame from which faculty were randomly sampled included all faculty in departments typically found in colleges of agriculture and life sciences within Land Grant Universities. In this work, we report results for a sub-sample that uses similar criteria for inclusion as used in 2005 and 2015. For 1989, we limited the original sample to eliminate non-professorial-rank, those in non-agricultural fields and off-campus extension units, resulting in 856 respondents. Note also that the 1989 wave of the survey does not include social science and agricultural engineering departments, which makes it not fully comparable to the following waves. We provide a robustness check of our results (see S1 Appendix) by restricting the 1995, 2005 and 2015 samples to be comparable to 1989. There are no discernible qualitative changes in the results from restricting our data to match the 1989 survey sample frame. In 1995, the sample of respondents was also limited to professorial-rank, and excluded individuals outside colleges of agriculture. The resulting sub-sample consists of 415 respondents.

Table 2 summarizes the final sample size for each year. It is important to note that sample size varies across the tables and figures presented above due to differential missing values across variables. Since restricting the sample to account for cross-missing data would compromising 10 to 20 percent of the data, we report in this paper results without restrictions. We, however, provide in S1 Appendix a robustness check with results generated after restricting to the sub-sample with non-missing response in all variables used. In that restricted data set all of our main effects remain qualitatively unchanged.

Response rates are calculated as the ratio between complete interviews and total interviews, excluding ineligible interviews. Response rates were 87.4 percent in 1989 (N = 920), 57.4 percent in 1995 (N = 656), 62.46 percent in 2005 (N = 1,433) and 33.54 percent in 2015 (N = 977). Response rates reported here include cross-section and panel individuals. In this study, we restrict the analysis to the cross-section sample in all years. For all years, samples are restricted following specific criteria. In 1989, individuals in extension units and non-agricultural

**Table 2. Total responses and number of observations used in this study, by wave.**

|  | Survey Responses | Response Rate (%) | Sub-sample used in this study |
|---|---|---|---|
| 1989 | 920 | 87.40 | 856 |
| 1995 | 656 | 57.40 | 415 |
| 2005 | 1,433 | 62.46 | 1,185 |
| 2015 | 977 | 33.54 | 711 |

Table 3. Sample/Faculty composition change across years.

|  | Women | Tenured | Applied Research |
|---|---|---|---|
| 1989 | 0.08 | 0.80 | 0.63 |
| 1995 | 0.08 | 0.86 ** | 0.69 ** |
| 2005 | 0.19 *** | 0.73 *** | 0.73 *** |
| 2015 | 0.27 *** | 0.75 * | 0.73 *** |
| Obs. | 3.158 | 3.090 | 2.986 |

"Tenured" is defined as faculty reporting appointment as "Associate" or "Full" professor. "Applied Research" is defined as faculty reporting 50 percent or more of research as 'applied'. Symbols indicate significance levels for the mean differences for each year. Baseline year is 1989.

*, < 0.1,

**, < 0.05,

*** < 0.01

departments (computer sciences, home economics, social scientists, agricultural engineers, and veterinary) were excluded. Reduced sample sums up to 856 in 1989. The same criteria is applied in 1995, and individuals in extension units and non-agricultural departments (computer science, home economics, veterinary) have been excluded.

Some tables and figures do not show data from 1989 and 1995 because either the questions were not asked in those years or the questions asked had different scaling than in 2005 and 2015. For Fig 5 the 1989 and 1995 surveys had similar questions, but with non-comparable scaling. The 1989 and 1995 surveys did not have detailed enough funding information for inclusion in Figs 6 and 7. For Table 1 and Fig 11, the relevant questions were not asked in 1989 and 1995.

The sample composition also changes across survey waves, explained by faculty change in LGUs over the decades. For instance, as shown in Table 3, the proportion of female faculty increases from 8 percent in 1989 to 27 percent in 2015, largely reflecting the historical documented increase representation of women in land grant universities [44]. We also see a decrease in the proportion of tenured faculty, from 80 to 75 percent, which is consistent with an ageing faculty cohort. There is also a substantial increase in the percent of faculty conducting some applied research. It is important to note that 80 percent of faculty has a research portfolio that mixes basic and applied research.

## Exploratory factor analysis

We conducted Exploratory Factor Analysis (EFA) to uncover potential underlying structured relationships (factors) in the data for all years related to attitudes and motivations. In the survey, respondents answered questions about their motivations when choosing a research topic, from which we selected a subset of 10 items common to all waves. Items were reported in a Likert Scale from 1 (not at all important) to 5(extremely important).

Using principal factor methods, our results suggest two underlying factors with eigenvalues 1.66 and 1.33. We retain two factors based on the "scree test". We apply an oblique promax rotation, keeping items with a factor loadings above 0.4. Results are displayed in Table 4. We chose principal factors so as to not impose strong distributional assumptions. Results are robust to the use of the maximum likelihood (ML) method [45].

**Table 4. Attitudes towards research.**

| | Factor Analysis Results | | Item average score, per year | | | |
|---|---|---|---|---|---|---|
| | *Factor 1* | *Factor 2* | **1989** | **1995** | **2005** | **2015** |
| Contribution to scientific theory | | 0.57 | 3.76 | 3.65 | 3.71 | **3.53** |
| Prob. pub. in professional journal | | 0.43 | 3.78 | 3.70 | **3.89** | 3.86 |
| Enjoy doing this kind of research | | 0.54 | 4.45 | 4.40 | **4.57** | **4.35** |
| Scientific Curiosity | | 0.62 | 4.09 | 4.04 | **4.22** | 4.14 |
| Availability of research facilities | | | 3.73 | **3.36** | **3.44** | **3.29** |
| Approval of colleagues | | | 2.51 | 2.49 | 2.48 | **2.30** |
| Importance to society | | | 3.77 | **3.96** | **4.28** | **4.09** |
| Marketability of final product | 0.46 | | 2.44 | 2.40 | 2.48 | **1.90** |
| Request made by clientele | 0.72 | | 2.56 | **2.92** | **3.03** | **2.75** |
| Feedback from extension personnel | 0.71 | | 2.24 | 2.36 | **2.52** | **2.35** |

This table shows factor loadings resulting from an exploratory factor analysis. We kept factors with loadings above 0.3. In bold are values statistically different from 1989, at a 10% significance level. We interpret factor 1 as "Extrinsic motivation" and factor 2 as "Intrinsic motivation".

## Supporting information

**S1 Fig. Comparing the proportion of faculty across bins of attitudes, by basic vs. applied research.** This figure displays the proportion of faculty across bins of intrinsic (A) and extrinsic (B) motivation. Proportions are calculated across two types of faculty research: Basic and Applied. In the survey, faculty inform the percentage of research time allocated to basic and applied research. Variable plotted in this graph refers to dummy variables. Faculty is classified as an "applied researcher" if at least 50 percent of their research time is allocated to applied research. N = 2,986.
(TIF)

**S2 Fig. Research time allocation across sub-activities by university ranking groups, in 2005 and 2015.** This figure displays the proportion of total time allocated to different types of activities by university ranking (A) Top 20 (B) Non-top 20. University ranking is based on the 2015 US News & World Report. For consistency, we use the 2015 ranking to classify universities in both 2005 and 2015 waves. Based on the general US News & World ranking, the top 20 best ranked LGUs are assigned as "top 20", the remaining are assigned as "non-top 20". This figure reports results restricted to 2005 and 2015 due to data availability. N = 1,741.
(TIF)

**S1 Table. Scholarly activities: Summary statistics across basic vs. applied researchers, by survey waves.** This table shows average for faculty scholarly activity outcomes. Averages are calculated across two types of faculty research: basic and applied. In the survey, faculty inform the percentage of research time allocated to basic and applied research. Faculty is classified as an "applied researcher" if at least 50 percent of their research time is allocated to applied research. Symbols refer to *p*-values for the mean differences *t*-test: *: 10%, **: 5%, ***:1%. N = 2,986.
(PDF)

**S1 Appendix. Robustness check.** In this appendix we show alternative methods of cutting down the sample to create consistency across the various surveys. The first is to make the

number of observations consistently the same across all estimates, while the second is to make the 1995, 2005, and 2015 data have similar sampling frames to the 1989 data.
(PDF)

**S1 File.**
(PDF)

**S2 File.**
(PDF)

**S1 Data.**
(ZIP)

## Acknowledgments

Thanks to Jessica Goldberger for her work on previous surveys, as well as Maria Isabel Agnes, Josh Alfonso, Vikas Gawai, and Jordan Van Rijn for data work on the 2015 survey.

## Author Contributions

**Conceptualization:** Bradford Barham, Jeremy Foltz, Ana Paula Melo.

**Data curation:** Bradford Barham, Jeremy Foltz, Ana Paula Melo.

**Formal analysis:** Bradford Barham, Jeremy Foltz, Ana Paula Melo.

**Funding acquisition:** Bradford Barham, Jeremy Foltz.

**Investigation:** Bradford Barham, Jeremy Foltz, Ana Paula Melo.

**Methodology:** Bradford Barham, Jeremy Foltz, Ana Paula Melo.

**Project administration:** Jeremy Foltz.

**Supervision:** Bradford Barham.

**Validation:** Bradford Barham, Jeremy Foltz, Ana Paula Melo.

**Visualization:** Bradford Barham, Jeremy Foltz, Ana Paula Melo.

**Writing – original draft:** Bradford Barham, Jeremy Foltz, Ana Paula Melo.

**Writing – review & editing:** Bradford Barham, Jeremy Foltz, Ana Paula Melo.

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
