## [Decision Letter · Decision Letter 0]

31 Aug 2021

PONE-D-21-19145

The enduring pursuit of public science at U.S. land-grant universities

PLOS ONE

Dear Dr. Foltz,

Thank you for submitting your manuscript to PLOS ONE. After careful consideration, we feel that it has merit but does not fully meet PLOS ONE’s publication criteria as it currently stands. Therefore, we invite you to submit a revised version of the manuscript that addresses the points raised during the review process.

Both reviewers are quite enthusiastic about the contribution your submission makes, and I share their assessment.  I believe, however, your analysis could be improved and its impact increased by addressing some or all of the points suggested by Reviewer #2.  In particular, I encourage you to consider the issue of stratification and differences in trends across the spectrum of land-grant institutions represented and to explore some of the correlations between measures in the data.  I will not need to send a revised version back to the reviewers, but I do ask that you consider the feasibility and desirability of the suggested additional analysis and either incorporate them in the revised manuscript or explain why you have chosen not to do so.

We look forward to receiving your revised manuscript.

Kind regards,

Joshua L Rosenbloom

Academic Editor

PLOS ONE

Journal Requirements:

Reviewers' comments:

Reviewer's Responses to Questions

**Comments to the Author**

1. Is the manuscript technically sound, and do the data support the conclusions?

Reviewer #1: Yes

Reviewer #2: Yes

2. Has the statistical analysis been performed appropriately and rigorously? 

Reviewer #1: Yes

Reviewer #2: Yes

3. Have the authors made all data underlying the findings in their manuscript fully available?

Reviewer #1: No

Reviewer #2: Yes

4. Is the manuscript presented in an intelligible fashion and written in standard English?

Reviewer #1: Yes

Reviewer #2: Yes

5. Review Comments to the Author

Reviewer #1: Overall, interesting results and comparison of faculty attitudes over time are noteworthy relative to the changing incentives.

The "methods" section describes the changing gender composition of faculty over time. It was a bit unclear whether the reported data in the main text adjust for these changes or not.

I didn't think figure 1 was necessary. Figure 4 was a bit confusing in terms of what the circles locations and sizes represented.

Reviewer #2: The authors present a fascinating analyses of the changes in the public science model. The article exploits four

waves of survey evidence from agricultural and life science faculty at the 52 major U.S. land-grant universities, spanning 1989-2015, to examine faculty attitudes/preferences, tenure and promotion criteria, output, and funding sources. The results show that attitudes toward scientific research have remained remarkably stable over twenty-five years in strongly favoring intrinsic and public science goals over commercial or extrinsic goals. We also demonstrate the faculty's positive attitudes toward science, an increased pressure to publish in top journals and secure increasingly competitive grants, as well as declining time for science.

There is much to like in this interesting, novel and clearly executed study. The authors are answering important question with new data. The findings are appropriately interpreted and the drafting of the paper is clear.

I suggest a few possible routes for improvement.

(1) There is little analysis of the relationships between the variables. While the authors demonstrate that attitudes change little it would be nice to know more about whether the changes they document Figures 6 and 7 affect faculty attitudes controlling for other factors. It would of interest to know whether declining research time - documented in figure 10 - affect faculty attitudes.

(2) I would be nice to see some analyses that look at the trends differently by university ranking. Many feel that their is increasing stratification within higher education and I wonder if your data reveal that. It would be interesting to know if research funding, time are being increasingly concentrated at elite institutions and how that might affect trends in attitudes by school rank.

(3) The paper is largely written as if commercial and basic science are substitutes. However recent research indicates that basic and applied research are complements (Akcigit, Hanley, Serrano-Velarde, 2021). In agriculture high level basic research does have positive spillovers to the commercial sector (Kantor and Whalley, 2019) and how the local commercial sector spills over to basic research (Sohn, 2021). This should be discussed.

(4) The paper has little to say about what exact commercial activities they expect from faculty members in this space. Many think of silicon valley software spinoffs as a positive externality from Stanford Computer Science faculty research. What are the likely spinoff technologies here?

References:

Akcigit, Hanley, Serrano-Velarde, 2021

https://academic.oup.com/restud/article/88/1/1/5922649?login=true

Kantor and Whalley, 2019

https://www.journals.uchicago.edu/doi/full/10.1086/701035?mobileUi=0

Sohn, 2021

https://pubsonline.informs.org/doi/full/10.1287/orsc.2020.1407

6. PLOS authors have the option to publish the peer review history of their article (what does this mean?). If published, this will include your full peer review and any attached files.

Reviewer #1: No

Reviewer #2: No

---

## [Author Response · Author response to Decision Letter 0]

25 Oct 2021

Comments from Reviewer #1:

The "methods" section describes the changing gender composition of faculty over time. It was a bit unclear whether the reported data in the main text adjust for these changes or not.

The reported data do not adjust for these changes. The changing gender composition of LGU faculty is the subject of on-going research by the PIs, which, while very interesting, is not easily included in this work.

I didn't think figure 1 was necessary. Figure 4 was a bit confusing in terms of what the circles locations and sizes represented.

We believe that figure 1 is necessary to the understanding of these processes and that explaining them without a figure would confuse the reader. We have kept it. We have revised the text around figure 4 in order to make clearer to the reader what the circles and their sizes represented.

Reviewer #2

(1) There is little analysis of the relationships between the variables. While the authors demonstrate that attitudes change little it would be nice to know more about whether the changes they document Figures 6 and 7 affect faculty attitudes controlling for other factors. It would of interest to know whether declining research time - documented in figure 10 - affect faculty attitudes.

We agree with the reviewer in principle but are not able to implement this as easily as they suggest would be possible. This work seeks to lay out the trends in these variables across time. Analyzing the relationships between variables as suggested requires careful attention to issues of identification, endogeneity, bad controls, and causality. As we have expounded in our discussion, that is an exceedingly difficult issue, beyond the scope of this work and likely impossible to do without accessing additional data. We leave it to future work to do this.

(2) I would be nice to see some analyses that look at the trends differently by university ranking. Many feel that their is increasing stratification within higher education and I wonder if your data reveal that. It would be interesting to know if research funding, time are being increasingly concentrated at elite institutions and how that might affect trends in attitudes by school rank.

We thank the reviewer for the suggestion and have implemented this in a number of places. We do indeed find some increasing stratification, especially with respect to research time. We have included this thought in a number of places in the text and included a new appendix figure that shows the major result of different levels of research time for top-20 LGU’s and bottom 30 LGUs.

(3) The paper is largely written as if commercial and basic science are substitutes. However recent research indicates that basic and applied research are complements (Akcigit, Hanley, Serrano-Velarde, 2021). In agriculture high level basic research does have positive spillovers to the commercial sector (Kantor and Whalley, 2019) and how the local commercial sector spills over to basic research (Sohn, 2021). This should be discussed.

We thank the reviewer for the additional references and have added 2 of the 3 of them to our discussion. We have also rewritten parts of the text so that it is clear we do not see commercialization and science as substitutes but as complementary activities. 

(4) The paper has little to say about what exact commercial activities they expect from faculty members in this space. Many think of silicon valley software spinoffs as a positive externality from Stanford Computer Science faculty research. What are the likely spinoff technologies here?

We have included some of this information in the introduction.

---

## [Editor Report · Decision Letter 1]

2 Nov 2021

The enduring pursuit of public science at U.S. land-grant universities

PONE-D-21-19145R1

Dear Dr. Foltz,

We’re pleased to inform you that your manuscript has been judged scientifically suitable for publication and will be formally accepted for publication once it meets all outstanding technical requirements.

Thank you for your responsiveness in considering the suggestions of the reviewers for improvement.  The revised version is clearly above the bar for publication, but I confess that I still find Figure 4 and its explanation somewhat opaque.  I assume that the position on the x-axis denotes the ranking of average scores across individuals, so it is not clear to me (still) what the size of the circles reflects.  You describe larger circles as reflecting higher relative average scores, but this doesn't much clarify things.  Perhaps this data would be more clearly conveyed with two graphs?

I hope you will consider ways to more clearly communicate this information, but you may proceed regardless.

Kind regards,

Joshua L Rosenbloom

Academic Editor

PLOS ONE
---

## [Editor Report · Acceptance letter]

12 Nov 2021

PONE-D-21-19145R1 

The enduring pursuit of public science at U.S. land-grant universities 

Dear Dr. Foltz:

I'm pleased to inform you that your manuscript has been deemed suitable for publication in PLOS ONE. Congratulations! Your manuscript is now with our production department. 

Kind regards, 

on behalf of

Dr. Joshua L Rosenbloom 

Academic Editor

PLOS ONE